# Attitude Tracking Algorithm Using GNSS Measurements from Short Baselines

**DOI:** 10.3390/s25216761

**Published:** 2025-11-05

**Authors:** Fedor Kapralov, Alexander Kozlov

**Affiliations:** Faculty of Mechanics and Mathematics, Lomonosov Moscow State University, 119991 Moscow, Russia; a.kozlov@navlab.ru

**Keywords:** global navigation satellite system (GNSS), attitude determination, integer ambiguity resolution (IAR)

## Abstract

**Highlights:**

**What are the main findings?**
We present a novel GNSS-based attitude tracking method for short baselines that significantly reduces computational complexity without compromising the accuracy achieved by established algorithms.We introduce an a priori error model for GNSS measurement errors that lends itself to a clear and intuitive geometric interpretation.

**What is the implication of the main finding?**
By improving computational efficiency in integer ambiguity resolution, the proposed method simplifies the implementation of real-time attitude tracking algorithms, especially in systems that combine GNSS with data from other sensors.

**Abstract:**

The paper addresses the problem of attitude determination using Global Navigation Satellite System (GNSS) measurements from multiple antennas mounted on a navigation platform. To achieve attitude determination by GNSS with typical accuracy down to tenths of a degree for one-meter baselines, GNSS phase measurements are employed. A key challenge with phase measurements is the presence of unknown integer ambiguities. Consequently, the attitude determination problem traditionally reduces to a nonlinear, non-convex optimization problem with integer constraints. No closed-form solution to this problem is known, and its real-time calculation is computationally intensive. Given an a priori initial attitude approximation, we propose a new algorithm for attitude tracking based on the reduction of the nonlinear orthogonality-constrained attitude estimation problem to a linear integer least squares problem, for which numerical methods are well known and computationally much less demanding. Additionally, a simple a priori model for GNSS measurement error variance is introduced, grounded on the geometry of satellite signal propagation through vacuum and the Earth’s atmosphere, providing a clear physical interpretation. Applying the algorithm to a real dataset collected from a quasi-static multi-antenna, multi-GNSS system with sub-meter baselines, we obtain promising results.

## 1. Introduction

Attitude determination problems arise in a wide range of applications and play a crucial role in navigation and vehicle control. It can be addressed by processing measurements from various sensor configurations. Modern Global Navigation Satellite Systems (GNSSs) provide an infrastructure for solving the attitude determination problem in open-sky scenarios. The primary advantages of using GNSSs for attitude determination include relatively high accuracy and precision, as well as lower cost, size, weight, and power consumption compared to other sensors offering similar accuracy.

It is well known that the attitude determination problem can be solved when the coordinates of two or more non-collinear vectors are known in coordinate systems whose relative attitude (orientation) is to be estimated. In satellite navigation, *baseline* vectors, defined as the vectors connecting phase centers of two GNSS antennas, are commonly used. The problem then reduces to determining the orientation of the *body reference frame*, which is fixed to the navigation object, with respect to the *navigation reference frame*. The latter is selected according to user requirements and may be an Earth-fixed frame or a locally level frame, such as the local geodetic frame (ENU, for East-North-Up) or the Earth-Centered Earth-Fixed (ECEF) frame. The coordinates of the baselines in the body reference frame are assumed to be known a priori; these may be determined through direct measurement or derived from the vehicle’s design specifications.

Carrier phase measurements can provide millimeter-level accuracy in baseline coordinates but contain unknown integer ambiguities. Resolving these integer ambiguities, commonly referred to as integer ambiguity resolution (IAR), often leads to an integer least squares (ILS) problem. A comprehensive literature survey on GNSS-based attitude determination methods, applications, and accuracy levels is presented by [1]. In addition, ref. [2] offers an excellent introduction to the theory of modern GNSS-based attitude determination methods and their applications.

One of the most traditional and straightforward approaches to GNSS-based attitude determination involves first estimating the baseline coordinates in the navigation reference frame by solving the ILS problem, followed by extracting the attitude matrix through solving the well-known Wahba’s problem [3]. This approach is based on the use of the *unconstrained* attitude model [4,5]. Hereafter, we refer to this method as the *baseline-based* method of attitude determination.

As detailed in Section 2.1, the GNSS-based attitude determination problem can be formulated as a least-squares optimization with integer constraints on the phase ambiguities and orthonormality constraints on the attitude matrix. This formulation is referred to as the optimization problem for the *orthonormality-constrained* attitude model [4]. Since a closed-form solution of this problem is unknown, different numerical methods should be applied to solve it. The problem becomes computationally demanding when high-rate, multi-antenna, multi-GNSS systems (GPS, GLONASS, Galileo, and BeiDou) are used in real-time applications.

Among existing techniques, the MC-LAMBDA algorithm [6] is currently regarded as one of the most robust and widely adopted methods for solving problems of this type. The MC-LAMBDA method uses state-of-the-art search techniques to resolve integer ambiguities in non-convex space due to orthonormal nonlinear constraints. The latter circumstance requires computationally intensive procedures [7].

Several approaches have been proposed that exploit simplifications of the orthonormality-constrained attitude model. Among these, the most classical is the baseline-based method, which assumes neither the orthonormality constraints nor the known baseline coordinates in the body reference frame at the IAR step. In addition, Teunissen [8] demonstrates that the orthonormality constraints can be replaced with affine constraints on the attitude matrix. This alternative formulation is known as the *affine-constrained* attitude model. These simplified attitude models enable linear ILS theory to be applied.

On the one hand, such methods reduce computational complexity by an order of magnitude compared to MC-LAMBDA. On the other hand, they provide a lower accuracy due to the partial loss of information at the problem formulation stage [4,5].

Also, alternative approaches have been proposed that strengthen the constraints inherent in the original problem statement. In [9], the authors proposed a constrained wrapped least-squares method (C-WLS) designed to reinforce the constraints inherent in the attitude determination problem. Namely, they utilize the fact that random errors of double-differenced carrier phase measurements are confined to a ±0.5-cycle interval. This additional assumption is almost always true and allows us to estimate the attitude matrix without the direct procedure of IAR. The method may perform better than MC-LAMBDA in terms of accuracy and computational efficiency in GNSS-challenging conditions when the multipath effect is strong.

In contrast to conventional attitude determination, which computes orientation at a single epoch solely from instantaneous GNSS measurements, attitude tracking relies on the continuous updating and refinement of the attitude estimate across successive measurement epochs. In this report, we propose a novel attitude model that enables the development of an attitude tracking algorithm with accuracy comparable to the MC-LAMBDA method but with significantly lower computational complexity. The proposed method is based on classical numerical techniques for solving the ILS problem and requires an initial attitude approximation.

The paper is organized as follows. In Section 2.1, we present the well-known optimization problem associated with the orthonormality-constrained attitude model. Section 2.2 introduces a model for the a priori variance of GNSS measurement noise, accompanied by a simple geometric interpretation. In Section 2.3, we propose a novel attitude model for GNSS-based attitude tracking along with the corresponding algorithm. In Section 3.1 and Section 3.2, simulations illustrate the scope of applicability of the proposed method and its typical accuracy. Finally, in Section 3.3 and Section 3.4, we evaluate the performance of the proposed algorithm through a quasi-static real experiment in which it is directly compared with the classic baseline-based attitude determination method across various multi-antenna GNSS setups.

## 2. Materials and Methods

### 2.1. Theoretical Background

Consider a multi-antenna GNSS setup in which B+1 GNSS antennas are rigidly mounted on a navigation platform, forming B≥2 non-collinear baselines. We assume that distances between the antennas do not exceed 50 m, so all atmospheric errors affecting GNSS measurements can be effectively mitigated by double differencing [10].

The mathematical single-epoch model for double differences of code and carrier phase measurements can be expressed in the following matrix form:(1)$$z_{\rho }^{i}=H_{\rho }^{i}Rb^{i}+\epsilon_{\rho }^{i},\;z_{\phi }^{i}=H_{\phi }^{i}Rb^{i}+a^{i}+\epsilon_{\phi }^{i},\;i=1,\ldots ,B,$$
where *i* is the index enumerating the baselines, and(2)$$z_{\rho }^{i},\;z_{\phi }^{i}\in R^{m_{i}},\;H_{\rho }^{i},\;H_{\phi }^{i}\in R^{m_{i}\times 3},\;R\in \mathrm{SO}\left(3\right),\;b^{i}\in R^{3},\;a^{i}\in Z^{m_{i}},\;m_{i}\in N.$$Here, the column vectors $z_{\rho }^{i}$ and $z_{\phi }^{i}$ represent the double differences of code and carrier phase GNSS measurements for the *i*-th baseline, respectively. The scalar $m_{i}$ denotes the number of double-differenced code or carrier phase measurements available from the *i*-th baseline. The matrices $H_{\rho }^{i}$ and $H_{\phi }^{i}$ are known design matrices, whose rows contain the differences of unit line-of-sight vectors from antennas to satellites. To calculate the design matrices, the code GNSS solution is commonly utilized. The matrix *R* is the sought orthogonal matrix with unit determinant. It describes an attitude of the body reference frame with respect to some navigation reference frame. Column vector $b^{i}$ contains the coordinates of the *i*-th baseline in the body reference frame. We assume that the baseline coordinates in the body reference frame are precisely known a priori. Column vector $a^{i}$ contains unknown integer ambiguities of carrier phase double differences. Random vectors $\epsilon_{\rho }^{i}$ and $\epsilon_{\phi }^{i}$ represent random errors of code and carrier phase measurements, respectively, with their a priori known covariance matrices: (3)$$E\left[\epsilon_{\rho }^{i}\right]=E\left[\epsilon_{\phi }^{i}\right]=0,\;E\left[\epsilon_{\rho }^{i}\epsilon_{\rho }^{i\;⊤}\right]=Q_{\rho }^{ii},\;E\left[\epsilon_{\phi }^{i}\epsilon_{\phi }^{i\;⊤}\right]=Q_{\phi }^{ii},$$
where E is the mathematical expectation. Hereinafter, 0 denotes zero matrices of an appropriate size. We believe that errors of code and carrier phase measurements are uncorrelated with each other: (4)$$E[\epsilon_{\rho }^{i}\epsilon_{\phi }^{k\;⊤}]=0,\;i,\;k=1,\ldots ,B.$$We may take into account the common antenna measurements being contained in double differences for all baselines as follows: (5)$$E\left[\epsilon_{\rho }^{i}\epsilon_{\rho }^{j\;⊤}\right]=Q_{\rho }^{ij},\;E\left[\epsilon_{\phi }^{i}\epsilon_{\phi }^{j\;⊤}\right]=Q_{\phi }^{ij},\;i\neq j.$$

Let $Mm_{1}+\ldots +m_{B}$ be the number of code (or carrier phase) measurements; thus, there are *M* code and *M* carrier-phase measurements, i.e., 2M in total. Combining the given code and carrier phase measurements at a single epoch, we define the following column vectors: (6)$$z_{\rho }:=\;\left[z_{\rho }^{1};\ldots ;z_{\rho }^{B}\right]\in R^{M},\;\epsilon_{\rho }:=\;\left[\epsilon_{\rho }^{1};\ldots ;\epsilon_{\rho }^{B}\right]\in R^{M},$$(7)$$z_{\phi }:=\;\left[z_{\phi }^{1};\ldots ;z_{\phi }^{B}\right]\in R^{M},\;\epsilon_{\phi }:=\;\left[\epsilon_{\phi }^{1};\ldots ;\epsilon_{\phi }^{B}\right]\in R^{M},$$(8)$$b:=\;\left[b^{1};\ldots ;b^{B}\right]\in R^{3B},\;a:=\;\left[a^{1};\ldots ;a^{B}\right]\in Z^{M},$$(9)$$z:=\;\left[z_{\rho };\;z_{\phi }\right]\in R^{2M},\;\epsilon :=\;\left[\epsilon_{\rho };\;\epsilon_{\phi }\right]\in R^{2M}.$$Hereinafter, we use ; as the notation for vertical matrix concatenation.

From expressions (Equation 3)–(Equation 5) for the covariance matrices, we define the covariance matrix for measurement errors ϵ as follows: (10)$$Q:=\;E[\epsilon \epsilon^{⊤}].$$

The problem of a GNSS-based attitude determination is reduced to a non-linear non-convex optimization problem with orthogonal constraints on the attitude matrix *R* and integer constraints on the phase ambiguities *a*: (11)$$\min_{R\in \mathrm{SO}\left(3\right),\;a\in Z^{M}}{∥z-B\left(R\right)-Aa∥}_{Q}^{2},$$
where ${∥\;\cdot \;∥}_{Q}^{2}:=\;{\left(\;\cdot \;\right)}^{⊤}Q^{-1}\left(\;\cdot \;\right)$. The matrices *B* and *A* are the known design matrices: (12)$$B(R):=\;\left[\mathrm{diag}\left(H_{\rho }^{1},\ldots ,H_{\rho }^{B}\right);\;\mathrm{diag}\left(H_{\phi }^{1},\ldots ,H_{\phi }^{B}\right)\right]\left(I_{B}⊗R\right)b,\;A:=\;\left[\begin{array}{c} 0\\ I_{M} \end{array}\right],$$
where *I* is the identity matrix of a given size, and ⊗ is the Kronecker product. The problem (Equation 11) is known as the optimization problem for the orthonormality-constrained attitude model.

### 2.2. Proposed a Priori Variance Model for Measurement Noise

Since the solution of the GNSS-based attitude determination problem (Equation 11) depends on the covariance (Equation 10), it is necessary to specify an exact a priori model used for our raw measurements. It is well known that propagation through the atmosphere introduces errors into GNSS measurements, so that, in general, for satellites with lower elevation over the horizon, the errors are larger. Although many models had been suggested in the past [11,12,13,14,15,16,17], from our experience of processing many different experimental datasets, we believe that using either model produces similar estimation results, and only ignoring the dependency entirely yields considerable degradation. In particular, Panetier, Bosser, and Khenchaf in [17] provide a proper comparison of different models.

This may change over time, but currently, we are using a model that is not listed in the above references but has a very straightforward physical interpretation, rather than being purely heuristic. We assume that a priori standard deviation of carrier phase measurement error for a particular radio signal is proportional to the distance traveled through the atmosphere. For satellite *s*, its measurement a priori standard deviation $\sigma_{s}$ takes the form given in (Equation 13), following from the geometry shown in Figure 1 (assuming spherical shape of the Earth and relatively thin atmosphere): (13)$$\begin{array}{c} \sigma_{s}∼\sigma_{0}\frac{d}{h_{a}}\approx \sigma_{0}\frac{R_{E}}{h_{a}}\left(\sqrt{\sin^{2}El+\frac{2h_{a}}{R_{E}}}-\sin El\right), \end{array}$$(14)$$\begin{array}{c} d=R_{E}\left(\sqrt{\sin^{2}El+2\frac{h_{a}}{R_{E}}+{\left(\frac{h_{a}}{R_{E}}\right)}^{2}}-\sin\;El\right). \end{array}$$Here, El denotes the elevation angle, defined as the angle between the horizon and the line-of-sight vector to the given satellite; $\sigma_{0}$ corresponds to a satellite at zenith (with El=90°); *d* represents the distance traveled by the signal through the actively interacting atmospheric layers. Their nominal total thickness is denoted by $h_{a}$, which we assume to be much smaller than the Earth’s radius $R_{E}$, so that $h_{a}/R_{E}\ll 1$ holds. The Formula (Equation 14) for *d* solves a quadratic equation produced by the law of cosines applied to the triangle OKC with ∠OKC=π/2+El in Figure 1.

Having fitted the Formula (Equation 13) to in situ carrier phase measurements back in 2009, we obtained reasonable values of(15)$$\sigma_{0}\approx 0.01\;\mathrm{cycles},\;R_{E}/h_{a}\approx 20\;\left(h_{a}\approx 320\;\mathrm{km}\right),$$
which we use to this day. The value of $h_{a}$ obtained from the fit in (Equation 15) conveniently happens to match the upper bound of the highest electron number density region in the ionosphere, being roughly at 300 km altitude according to conventional nomenclature (Introduction [18]). Please note that for different combinations of GNSS antennas and receivers, $\sigma_{0}$ may vary from approximately 0.005 to 0.03 cycles, but the dimensionless weighting factor depending on the elevation angle retains its form.

### 2.3. Proposed Attitude Tracking Method

The proposed method of attitude tracking using carrier phase measurements is based on the idea of simplification of the problem (Equation 11). However, rather than loosening or adding new constraints, we will follow the ideology of reducing the original problem under some frequently satisfied conditions to a simpler problem, the numerical solution of which is simpler and well understood.

Let the attitude at a fixed epoch be approximately estimated. If *R* is a true sought attitude matrix, then for its approximate estimate $R^{\prime }$, the following approximation holds: (16)$$R=\left(I_{3}-{\left[\delta \right]}_{\times }\right)R^{\prime },\;∥\delta ∥\ll 1,$$
where $\delta :={\left[\delta_{1},\;\delta_{2},\;\delta_{3}\right]}^{⊤}$ is the Euler vector corresponding to a small rotation angle, written in the navigation frame, and ${\left[\;\cdot \;\right]}_{\times }$ is the cross-product matrix defined as follows: (17)$${\left[\delta \right]}_{\times }:=\;\left[\begin{array}{ccc} 0 & -\delta_{3} & +\delta_{2}\\ +\delta_{3} & 0 & -\delta_{1}\\ -\delta_{2} & +\delta_{1} & 0 \end{array}\right].$$

The relation given in (Equation 16) is a linear approximation of Rodrigues’ rotation formula. The approximate estimate of the true attitude matrix, $R^{\prime }$, can be obtained either from non-GNSS sensor measurements, such as those provided by inertial measurement units (IMUs), magnetometers, sun sensors, or star trackers, or through GNSS-based algorithms for instantaneous attitude determination. Methods for obtaining $R^{\prime }$ are discussed in more detail later in this section.

Substituting (Equation 16) into the second equation of (Equation 1), we obtain the following expressions: (18)$$z_{\phi }^{i}-H_{\phi }^{i}R^{\prime }b^{i}=H_{\phi }^{i}{\left[R^{\prime }b^{i}\right]}_{\times }\delta +a^{i}+\epsilon_{\phi }^{i},\;i=1,\ldots ,B.$$

Let us define the phase double-differenced measurements $\zeta_{\phi }$ corrected for known a priori attitude, and the design matrix $H_{\phi }$: (19)$$\zeta_{\phi }:=\;\left[\begin{array}{c} z_{\phi }^{1}-H_{\phi }^{1}R^{\prime }b^{1}\\ \vdots \\ z_{\phi }^{B}-H_{\phi }^{B}R^{\prime }b^{B} \end{array}\right]\in R^{M},\;H_{\phi }:=\;\left[\begin{array}{c} H_{\phi }^{1}{\left[R^{\prime }b^{1}\right]}_{\times }\\ \vdots \\ H_{\phi }^{B}{\left[R^{\prime }b^{B}\right]}_{\times } \end{array}\right]\in R^{M\times 3},$$
then, (Equation 18) may be written in the matrix form: (20)$$\zeta_{\phi }=H_{\phi }\delta +a+\epsilon_{\phi }.$$

The system (Equation 20) contains 3+M unknowns and *M* equations, but, instead of involving *M* code measurements $z_{\rho }$, we augment the phase-based model with three equations for a priori attitude error δ as follows: (21)$$\zeta_{\delta }:=\;0,\;\zeta_{\delta }=\delta +\epsilon_{\delta },\;E\left[\epsilon_{\delta }\right]=0,\;E\left[\epsilon_{\delta }\epsilon_{\delta }^{⊤}\right]=\sigma_{\delta }^{2}I_{3},\;E\left[\epsilon_{\delta }\epsilon_{\phi }^{⊤}\right]=0,$$
where $\epsilon_{\delta }$ is the random error of attitude increment measurements $\zeta_{\delta }$ with zero mean and a diagonal covariance matrix with a priori known standard deviations $\sigma_{\delta }$. Equation (Equation 21) is justified, as it relies on the assumption that δ is small, as in (Equation 16).

Combining (Equation 20) and (Equation 21), we obtain the system: (22)$$\left[\begin{array}{c} \zeta_{\delta }\\ \zeta_{\phi } \end{array}\right]=\left[\begin{array}{cc} I_{3} & 0\\ H_{\phi } & I_{M} \end{array}\right]\left[\begin{array}{c} \delta \\ a \end{array}\right]+\left[\begin{array}{c} \epsilon_{\delta }\\ \epsilon_{\phi } \end{array}\right].$$

Introducing the matrices as follows: (23)$$\zeta :=\;\left[\begin{array}{c} \zeta_{\delta }\\ \zeta_{\phi } \end{array}\right]\in R^{3+M},\;D:=\;\left[\begin{array}{c} I_{3}\\ H_{\phi } \end{array}\right]\in R^{(3+M)\times 3},\;\varepsilon :=\;\left[\begin{array}{c} \epsilon_{\delta }\\ \epsilon_{\phi } \end{array}\right]\in R^{3+M},\;P:=\;E\left[\varepsilon \varepsilon^{⊤}\right],$$
we obtain the mixed integer least squares (ILS) problem: (24)$$\min_{\delta \in R^{3},\;a\in Z^{M}}{∥\zeta -D\delta -Aa∥}_{P}^{2}.$$

Although the integer least-squares problem is NP-hard in general, there exist algorithms that often achieve polynomial expected complexity in practical scenarios [19]. Owing to the possibility of orthogonal decomposition of the objective function, the ILS problem can be solved in three steps [20].

First, a so-called *float solution* and its error covariance are obtained by ignoring the integer constraints for the phase ambiguities and applying the least squares. Second, the float solution serves as an initial estimate for ambiguity resolution in the integer domain. Among widely used approaches for this step, there is the LAMBDA method [10,21,22] and its optimized variant, MLAMBDA [23,24]. These methods employ optimized search procedures and leverage decorrelating ambiguity transformations that flatten the spectrum of conditional variances of ambiguities and thereby accelerate the search for the numerical solution. Finally, the resolved integer ambiguities are used to estimate the sought mixed ILS solution.

Let us discuss the motivation for incorporating measurements of attitude error $\zeta_{\delta }$ instead of double-differenced code measurements $z_{\rho }$, as in problem (Equation 11). The principal advantage of this approach is the computational effort being reduced in resolving the ILS problem.

First, the proposed problem (Equation 24) contains 3+M equations as opposed to the problem (Equation 11), where the number of equations is 2M. We note that the design matrix of (Equation 22) is square and has full rank; therefore, a unique float solution always exists. It is straightforward to verify that $\left[0_{3\times 1};\;\zeta_{\phi }\right]$ is the float solution of (Equation 24). Hence, the float solution is trivial, and no additional computations are required to obtain it, in contrast to methods that use code measurements $z_{\rho }$ in the problem formulation.

Second, it is known that the accuracy of the float solution in problem (Equation 11) is driven by the precision of the code measurements [6]. Therefore, if an approximate attitude is known with accuracy surpassing that achievable from typical double-differenced code measurements, then solving the proposed problem (Equation 24) yields a more precise float solution. Consequently, this enables us to use more computationally efficient ILS resolution via a more effective search process, such as the LAMBDA method.

Indeed, the last statement can be illustrated by the following rough estimate. Assuming $\sigma_{\delta }\approx 2$°, $\max_{i}∥b^{i}∥\approx 1\;m$, and typical double-differenced code measurements accuracy $\sigma_{\rho }\approx 40\;\mathrm{cm}$, we observe that $\sigma_{\rho }\gg \sigma_{\delta }\cdot \max_{i}∥b^{i}∥\approx 4\;\mathrm{cm}$. Hence, the proposed problem (Equation 24) produces a float solution approximately an order of magnitude more accurate in the case of short baselines and an approximately known attitude.

The proposed problem (Equation 24) yields a more accurate attitude solution than the problems based on the unconstrained or affine-constrained attitude models because it follows directly from the orthonormality-constrained attitude model. Meanwhile, the proposed linear problem does not require computationally intensive numerical methods such as MC-LAMBDA, which deals with non-linear constraints and a non-convex search space, making it prohibitively time-consuming for some real-time applications. In [7], the authors demonstrate that the average computation time of MC-LAMBDA in a real experiment may be almost two orders of magnitude greater than that required to resolve the ILS problem via LAMBDA. However, a fair comparison of the computation times of MC-LAMBDA and LAMBDA is not possible, as these methods are applied to different problem formulations and yield solutions with different accuracies. In Section 3.4.4, we present a similar comparison of the computation speeds of the proposed method based on (Equation 24) and the MC-LAMBDA method, using real data.

The key difference between our method based on the problem (Equation 24) and the MC-LAMBDA method is that the solution of (Equation 24) enforces the orthogonality constraints on the attitude matrix up to first-order terms in ∥δ∥ throughout all stages of the calculation, including both the float estimate and the IAR stage. In contrast, the MC-LAMBDA method discards the orthonormality constraints at the float estimation step. Additionally, problem (Equation 24) involves the minimal possible number of unknown attitude parameters, with $\delta \in R^{3}$.

The principal limitation of the proposed formulation lies in its requirement for an initial approximation of the attitude, naturally raising the question: how can the approximate attitude matrix $R^{\prime }$ be obtained in practice?

Two primary categories of methods for resolving the initial attitude estimation problem exist. The first encompasses approaches that incorporate auxiliary information beyond carrier phase measurements, whether additional GNSS-based data or external sensor data.

For instance, in spacecraft applications, high-accuracy star sensors can be employed to determine attitude [25], or attitude estimates within a few degrees can be achieved using signal-to-noise ratio (SNR) measurements from GNSS antennas [26], or a magnetometer-only algorithm based on Kalman filtering can provide ≈1.5° accuracy in attitude [27].

The specific choice of sensors is determined by the design objectives and application requirements of the navigation system. A widely adopted sensor combination employs an IMU with GNSS data [28,29,30,31]. It should be noted that the unknown parameter of angular increment, δ, represents an attitude error in the Kalman filter for IMU/GNSS integration, where $R^{\prime }$ denotes an attitude matrix obtained from an aided solution.

The second category comprises single-epoch GNSS-based attitude determination algorithms. It may be the MC-LAMBDA method, the C-WLS method, or low-complexity algorithms which disregard the constraint R∈SO(3) while still operating within the framework of mixed ILS theory; see, for example, [4,5,8]. By employing these methods to obtain an approximate attitude matrix, the proposed problem can be adapted for an instantaneous attitude determination algorithm.

Another approach is to solve the proposed problem while taking attitude history into account. Namely, we propose the following Algorithm 1.
**Algorithm 1:** The proposed algorithm of attitude trackingObtain an estimate of initial attitude matrix $\tilde{R}\left(t_{0}\right)$.Solving problem (Equation 24) at epoch $t_{k}$ with $R^{\prime }=\tilde{R}\left(t_{k-1}\right)$ and $R=\tilde{R}\left(t_{k}\right)$, obtain the estimate of Euler vector $\tilde{\delta }\left(t_{k}\right)$ and its accuracy given by a corresponding conditional covariance.Use $\tilde{\delta }\left(t_{k}\right)$ to obtain $\tilde{R}\left(t_{k}\right)$ using Rodrigues’ rotation formula applied to matrix $\tilde{R}\left(t_{k-1}\right)$.

Here $\tilde{\;\cdot \;}$ denotes an estimate of a quantity, the index *k* enumerates GNSS epochs, and $\tilde{R}\left(t_{k}\right)$ is the sought attitude matrix at the *k*-th epoch. Parameter δ can be interpreted as an attitude change between two consecutive GNSS epochs.

The first initialization step of the algorithm can be performed using other GNSS-based algorithms of instantaneous attitude determination or external measurements as discussed earlier.

The second step is essentially a straightforward prediction step, in which the attitude from the previous epoch is adopted as the attitude estimate for the current epoch. The overall accuracy and scope of applicability of the proposed algorithm depend on the precision of the prediction step. Naturally, the algorithm is expected to perform well in static conditions or for high-rate GNSS observations. In Section 3.4, we demonstrate that the proposed approach also delivers robust performance under low-dynamic angular motion in a real experiment.

It is important to note that such a simple prediction method is appropriate for many high-rate GNSS applications. In fact, the key assumption (Equation 16) holds when the GNSS sampling rate exceeds, for example, 20 Hz and the angular rate remains below approximately 120°/s. For reference, this angular rate threshold exceeds the upper bound of typical angular velocities encountered in most passenger aircraft.

Finally, it is important to note that the concept of linearization underlying the proposed algorithm is well established. For example, ref. [32] investigated the Linearized Constrained LAMBDA (LC-LAMBDA) method, which relies on linearizing the baseline attitude model subject to a nonlinear baseline length constraint.In the present work, we build upon ideas from the field of integrated navigation, as detailed in [33,34,35].

## 3. Results

### 3.1. Simulation Setup

An analytical study of numerical methods that directly address nonlinear problems with integer constraints, such as (Equation 11), is highly non-trivial. To assess the applicability of the proposed ILS formulation (Equation 24) and to compare its accuracy with that of the orthonormality-constrained attitude model (Equation 11), we perform numerical simulations encompassing both static and dynamic conditions. Static simulations evaluate the potential accuracy of the proposed problem formulation under fixed baseline conditions without temporal variation, serving as a benchmark for intrinsic estimation capability. The dynamic simulations apply the proposed algorithm under a rotation, thereby examining its robustness and performance in realistic operational environments.

To numerically solve the proposed ILS problem, we implement the MLAMBDA algorithm as described in [23,24]. For the problem (Equation 11), we employ the MC-LAMBDA method [6,20]. The classic MC-LAMBDA method exploits both code and carrier-phase measurements, whereas the proposed formulation (Equation 24) relies on a priori attitude information combined with carrier-phase measurements. To simplify the comparison of methods, the standard deviation of carrier-phase noise is fixed at a single typical value.

We consider a fixed satellite geometry with five satellites operating at a single frequency and four-antenna configurations, where three antennas form a right triangle with the fourth (base) antenna positioned at its centroid. Measurements from different satellites are assumed to have equal accuracy within each observable type. The antenna configurations differ only by their inter-antenna baseline lengths. When the body and navigation frames are aligned ($R=I_{3}$), all antennas lie within the horizontal plane. Table 1 summarizes the parameters of the numerical simulations.

Note that although chosen baseline configurations provide sufficient measurements for GNSS-based attitude determination, they are known to yield relatively poor accuracy among four-antenna configurations [20,35]. This claim is also demonstrated in Figure 2.

#### 3.1.1. Static Single-Epoch Conditions

For each combination of variable parameters—including baseline lengths and noise levels of code and a priori attitude measurements—solutions of problems (Equation 11) and (Equation 24) are generated using randomly drawn attitude matrices *R*, integer carrier-phase ambiguities *a*, and zero-mean normally distributed measurement errors $\epsilon_{\rho }$, $\epsilon_{\phi }$, and $\epsilon_{\delta }$. Fixed sets of parameters establish the experimental conditions, while the variable parameters are systematically swept across simulations. Each method is evaluated using $10^{4}$ samples for each combination of variable parameters.

#### 3.1.2. Dynamic Conditions

To estimate the region of validity for the proposed attitude tracking algorithm, simulations are conducted in which the GNSS antenna setup uniformly rotates about a vertical axis passing through the base antenna at various constant angular velocities. This type of motion may violate two assumptions of the proposed attitude tracking algorithm: first, that the attitude increment is zero-mean (Equation 21); and second, that the attitude increment remains small within a GNSS step (Equation 16). Similar to the previous section, we randomly generate integer carrier-phase ambiguities *a* and zero-mean normally distributed carrier-phase measurement errors $\epsilon_{\phi }$ at each epoch. The sampling frequency, typical for high-rate GNSS setups, is fixed at 20 Hz. The algorithm’s initialization error is set to zero. Table 1 and Table 2 summarize the key parameters used in the dynamic experiments involving uniform rotations.

### 3.2. Simulation Results

#### 3.2.1. Static Single-Epoch Conditions

Simulation results obtained with the setup described above are presented in Figure 3, where the solution accuracy of the proposed ILS problem is compared with that of MC-LAMBDA in terms of success rate and mean circular attitude error under varying baseline lengths and measurement noise levels.

As shown in Figure 3a, the IAR success rate of MC-LAMBDA is nearly insensitive to baseline length, in contrast to the proposed method. For the considered baseline configurations, the success rates of MC-LAMBDA and the proposed method are comparable and exceed 99.9% when the a priori attitude error satisfies $\sigma_{\delta }\leq 2.5$° and the standard deviation of code noise does not exceed 6 cm.

Figure 3b further shows that, for short sub-meter baselines, the proposed problem formulation achieves a notable accuracy improvement over MC-LAMBDA when $\sigma_{\delta }\leq 5$°, and the code noise level exceeds 6 cm. Specifically, under these conditions, the mean attitude error of the proposed problem solution does not exceed 0.7°, whereas the minimal mean error of MC-LAMBDA is approximately 1.5° and is reached on a 1 m baseline.

For MC-LAMBDA, increasing the baseline length consistently improves mean attitude accuracy regardless of the code noise level. The situation is more complex for the proposed problem formulation. For the considered sub-meter antenna configurations, increasing the baseline length improves its mean attitude accuracy when $\sigma_{\delta }\leq 4$°; however, when $\sigma_{\delta }>4$°, accuracy deteriorates as the baseline length increases. Thus, for configurations with sub-meter baseline lengths, the validity range of this formulation can be conservatively estimated as $\sigma_{\delta }\leq 3$°. This estimate aligns with the key assumption (Equation 16) underlying the problem. Indeed, assuming $\delta ∼N(0,\;\sigma_{\delta }^{2}I_{3})$, it follows that P(∥δ∥≤7°)>85% provided $\sigma_{\delta }\leq 3$°.

Both MC-LAMBDA and the proposed problem formulation have their own tuning parameters, and depending on the a priori accuracy of the measurements, each method may outperform the other under different conditions. It is clear that for short baselines, the proposed formulation can offer significant advantages over MC-LAMBDA, while MC-LAMBDA remains applicable to a broader range of baseline configurations.

#### 3.2.2. Dynamic Conditions

Figure 4 shows the estimated applicability scope of the proposed algorithm, which is obtained as follows. For varying baseline lengths and a priori noise levels of the attitude increment, $\sigma_{\delta }$, under different angular velocities, we determine, for each fixed angular velocity, the maximum baseline length for which the success rate of IAR is 100% when $\sigma_{\delta }\approx 3$°. This condition implies that there exists a value of tuning parameter, $\sigma_{\delta }$, yielding high accuracy, as supported by the results in Section 3.2.1, even under dynamic conditions.

As expected, the smaller the baseline length is, the more stable the algorithm becomes at large angular velocities. In our simulation, for a sub-meter baseline, our attitude tracking algorithm can handle angular rates up to 150°/s, and for half-meter baselines, up to 250°/s. Note that although the results are obtained for a selected configuration, results for other GNSS sampling frequencies can be obtained by scaling Figure 4b along the horizontal axis, because angular velocity is proportional to the GNSS sampling rate.

### 3.3. Experimental Setup

On 2 August 2023, a multi-antenna, multi-GNSS quasi-static experiment took place in Moscow under relatively favorable urban conditions. Five GNSS antennas were installed on a rotatable wooden platform. The distance between any two antennas was less than one meter, forming ten short baselines from the five antennas. The platform was oriented roughly horizontally. Satellite measurements were recorded at seven static positions, achieved by performing three clockwise rotations of 90° each around the vertical axis, followed by three counterclockwise rotations of the same magnitude. Each static position lasted for 20 min, with the entire experiment being approximately 2.5-h long.

We used four dual-band antennas of aviation grade, and the dual-band survey antenna, NovAtel GNSS-804 (NovAtel Inc., Calgary, AB, Canada), with high phase center stability within 2 mm. Antennas were connected to one dual-antenna receiver and three single-antenna receivers. A detailed scheme of the GNSS hardware connection is given in Table 3.

All receivers processed GPS and GLONASS signals in two frequency bands, while only two NovAtel receivers processed Galileo and BeiDou signals. NovAtel receivers recorded GNSS measurements at a rate of 20 Hz, whereas Javad receivers recorded at 10 Hz. To enable joint processing of measurements from both receiver types, the NovAtel data was decimated to 10 Hz, and all measurements are then utilized at this common sampling rate.

All antennas of the same model were installed with the same orientation relative to the body frame associated with the board. An overview of the multi-antenna GNSS setup and experiment conditions is provided in Figure 5.

The median PDOP values lie in the range of 1.5–2.5 for single-constellation GNSS configurations across different antennas. For the multi-constellation GNSS configuration, the median PDOP is approximately 0.7 for antennas $A_{1}$, $A_{2}$, and $A_{5}$.

### 3.4. Experimental Results

In this section, we analyze two kinds of multi-antenna GNSS attitude solutions:A three-antenna system utilizing only GPS signals at the L1 frequency, hereinafter referred as to Configuration *A*.A five-antenna system employing all available GNSS constellations (GPS, GLONASS, Galileo, and BeiDou) in both L1 and L2 frequency bands (Configuration *B*).

The first configuration represents a constrained, single-frequency GNSS setup, typically regarded as the most challenging scenario for GNSS-based attitude determination, albeit being available with the cheapest carrier phase-capable GNSS receivers. The second configuration demonstrates a modern, multi-antenna, multi-GNSS approach, exploiting signals from multiple constellations and frequency bands to provide the most accurate solution achievable with the proposed algorithm. The accuracy achievable with other multi-antenna GNSS setups is expected to fall between the two.

In both configurations, we select the sole survey-class antenna in the setup, $A_{5}$, as the base antenna. For Configuration *A*, we analyze solutions obtained using measurements from antennas $A_{1}$, $A_{2}$, and $A_{5}$. For Configuration *B*, we analyze solutions obtained using measurements from all five antennas. It is important to note that GLONASS measurements from antennas $A_{3}$ and $A_{4}$ are excluded in the Configuration *B* processing because inter-frequency bias differences between the heterogeneous receiver models prevent their reliable use in the attitude solution [36].

In this research, numerical results are obtained using system-specific double differencing, so that for each GNSS constellation, a reference satellite from that same constellation is selected. Experimental results [37] indicate that inter-system and system-specific double differencing of multi-GNSS measurements yield comparable attitude accuracy under favorable GNSS conditions, i.e., when a sufficient number of satellites are visible in an open-sky environment.

For each configuration, we analyze two types of attitude solutions. The first type is the solution obtained using the baseline-based method, where baseline vectors are estimated in the navigation reference frame, and the attitude matrix is then extracted by solving the Wahba problem. The second type comprises solutions obtained by applying the proposed Algorithm 1 from Section 2.3.

The longitudinal axis of the body frame is approximately aligned with the baseline $A_{1}A_{2}$, the second (normal) axis is roughly perpendicular to the plane formed by points $A_{1}$, $A_{2}$, $A_{3}$, and $A_{4}$, and the transverse axis is approximately directed along the baseline $A_{1}A_{4}$. The specific numerical parameters for experimental data processing—such as a priori standard deviations of GNSS measurements, baseline geometry, and others—are listed in Table A1. For newer receivers and antennas ($A_{1,\;2,\;5}$) we experimentally established a priori value of $\sigma_{0}$ to be 0.01 cycles according to (Equation 15), while for older models ($A_{3,\;4}$) we set $\sigma_{0}$ to twice that value to account for their higher noise component.

We represent the attitude of the body frame using three Euler angles, α:=(γ,θ,ψ) of roll, pitch, and yaw, respectively. In the remainder of this section, we analyze key aspects of the attitude solution quality: high-frequency noise, low-frequency “slow” components of attitude errors, and fraction of outliers. Additionally, we present brief results on computation times for different methods.

#### 3.4.1. Error Noise Analysis

To estimate the stochastic noise component of the attitude error, we extract filtered versions of the solutions using a high-pass finite impulse response (FIR) filter, denoted as F, with a cut-off frequency of 1 Hz, applied to time intervals corresponding to static periods in the experiment. During these static intervals, all solutions except one contain almost no outliers, so the filtered solutions are free from significant disturbances. However, the solution from Configuration *A* using the traditional baseline-based method exhibits many outliers (≈30% of epochs); therefore, we manually exclude these outliers before applying the filter F to avoid misinterpretation of accuracy metrics. Noise errors are characterized by the standard deviation of the following quantity: $\alpha_{\mathrm{HF}}:=F\left(\alpha \right)$. Typical noise levels of the attitude errors are presented in Table 4.

The proposed method reduces the error noise of roll, pitch, and yaw by approximately 40%, 20–50%, and 40%, respectively, compared to the baseline-based method. Using Configuration *B* instead of Configuration *A* reduces the noise in all angles by a factor of approximately 2–3.

#### 3.4.2. Outlier Analysis

To detect outliers in attitude angles during both static and rotation intervals, we first obtain a “reference” solution as follows. We consider the solution produced by the proposed algorithm for Configuration *B* as the most accurate. This solution is then processed using a two-pass moving median filter (applied forward and backward) with a window width of 40 s for pitch and roll angles, and 4 s for yaw. The order-of-magnitude difference in window widths reflects the difference in the typical rates of change of the corresponding angles observed in the experiment.

We denote the reference solution as $\alpha_{\mathrm{ref}}$. Let us define an attitude solution error estimate as follows: $\Delta \alpha :=\;\alpha -\alpha_{\mathrm{ref}}$. If the following criterion is satisfied, (25)$$|\Delta \alpha |:=\;|\alpha -\alpha_{\mathrm{ref}}|\;\geq 5\sigma \left(\alpha_{\mathrm{HF}}\right),$$
then the corresponding attitude angle α is considered an outlier. The fraction of epochs containing outliers in the attitude solutions is presented in Table 5.

Fractions of epochs with outliers in at least one attitude angle for Configuration *A* are 27.8% for the baseline-based method and 19.7% for the proposed method. For Configuration *B*, the fraction of epochs affected by outliers in at least one attitude angle is 4.7% and 3.2% for the baseline and proposed methods, respectively.

Although the differences in outlier fractions may seem modest due to the predominantly static nature of the experiment (with approximately 92.5% static epochs), the proposed algorithm performs significantly better under more challenging conditions. This occurs during board rotations, when part of the sky is blocked and the GNSS constellation geometry changes substantially. We illustrate this phenomenon for Configuration *B* in Figure 6. Figure 6a shows outliers in the baseline-based attitude solution occurring during a significant change in GNSS conditions around the epoch number $7.35\times 10^{4}$. A qualitatively similar pattern is observed in other intervals involving rotations. Figure 6b presents distributions of absolute errors compiled over all six rotation intervals for Configuration *B*.

Outliers in attitude solutions can often result from incorrect integer ambiguity resolution. However, for both configurations, the bootstrapped lower bounds of the ambiguity resolution success rate [38] achieved by the proposed algorithm exceed 99.999% at every epoch. This result leads us to believe that ambiguity resolution is not the main cause of attitude solution outliers for the proposed method.

Yaw angle estimates are the most accurate because they are mainly affected by errors in the horizontal components of the carrier phase measurements, which are generally less noisy than the vertical components. The difference in accuracy between pitch and roll can be attributed to the baseline geometry: the longitudinal baselines $A_{1}A_{2}$ and $A_{3}A_{4}$ are approximately three times longer than the transverse baselines $A_{1}A_{4}$ and $A_{2}A_{3}$, as shown in Figure 5a.

#### 3.4.3. Solution Stability Analysis

To analyze stability of attitude solutions in statics, we define the low-frequency components of attitude errors as follows: $\alpha_{\mathrm{LF}}:=G\left(\overset{\circ }{\alpha }\right)$, where G is a two-pass moving average filter (applied forward and backward) with a 10-s window, and $\overset{\circ }{\cdot }$ denotes a mean-centered value, with mean taken over different static intervals. Low-frequency errors in attitude estimation may be caused by multipath effects, atmospheric variations, small deviations in satellite orbits, or errors in the mathematical models. We characterize the range of slowly varying stochastic attitude errors using the percentile-based metric $P_{99.7}\left(\right|\alpha_{\mathrm{LF}}\left|\right)$, summarized in Table 6. This metric is chosen for its robustness to the choice of smoothing filter G. As shown in Table 5, a significant fraction of outliers occur in the baseline-based solutions for Configuration *A*. Therefore, for this solution, we first remove all outliers before applying the smoothing filter G.

For Configuration *A*, the proposed method reduces the range of low-frequency errors in roll, pitch, and yaw by approximately 40%, 30%, and 60%, respectively, compared with the baseline-based attitude model. For Configuration *B*, it reduces the range of low-frequency errors in roll and yaw by about 10% and 40%, respectively, but increases the pitch error range by approximately 10%. Overall, the proposed method achieves roll and pitch accuracy that is better than or comparable to that of the baseline-based method, while significantly improving the stability of the yaw angle.

Figure 7 illustrates a comparison of low-frequency angle errors for different attitude estimation methods using Configuration *B*.

As shown in Figure 7a, the range of low-frequency roll errors during the static period decreases from approximately 2.8° using the baseline-based method to 2.3° with the proposed method, indicating a notable improvement in solution stability.

Figure 7b presents the cumulative distribution functions of absolute low-frequency errors for all attitude angles across all static intervals. The proposed method achieves noticeable stability improvements in roll and yaw angles, while its effect on pitch stability is minimal.

#### 3.4.4. Computational Time Comparison

To compare the typical computational complexity of the proposed method and MC-LAMBDA, their computation times are evaluated for Configuration A. Table 7 presents the median computation times along with the corresponding median absolute deviations (MAD). The median is reported instead of the mean to minimize the influence of occasional epochs when MC-LAMBDA exhibits significantly prolonged runtimes. We do not present statistics for the baseline-based method due to its near-complete identity with the proposed method.

The computation times of the two methods were measured using MATLAB 2020b on a Windows 10 system equipped with an Intel Core i7 8565U processor to assess their relative performance. It should be noted that more advanced implementations of MC-LAMBDA [7] are likely to exhibit improved performance compared to the classic implementation [6,20] employed in this study. Nevertheless, the proposed method achieves computational speeds approximately two orders of magnitude faster than MC-LAMBDA, which aligns well with the results reported in [7] (p. 508).

## 4. Discussion

The proposed attitude tracking algorithm offers significant advantages over conventional methods, such as MC-LAMBDA and the baseline-based approach, by achieving markedly higher computational efficiency and eliminating the need for explicit GNSS code measurements. It also demonstrates superior accuracy across a wide range of high-rate multi-antenna GNSS configurations with short baselines; for example, in setups comprising four antennas spaced by approximately half a meter and operating at a sampling rate of 20 Hz under angular velocities up to 250°/s. However, the algorithm requires an initial attitude estimate, and its applicability and accuracy are constrained by the geometric configuration of the GNSS antennas and by the magnitude of the attitude increment within each GNSS sampling step. The latter factor, in turn, depends on the GNSS sampling rate and the platform’s angular velocity. Based on the obtained results, the proposed method is most suitable for high-rate multi-antenna GNSS systems operating either in low-dynamic conditions with angular rates up to 60°/s and baselines on the order of a few meters, or in active angular motions up to 150–250°/s with sub-meter baselines. Furthermore, as GNSS sampling rates continue to increase with modern receiver technology, the applicability range and robustness of the presented algorithm are expected to improve accordingly.

Using a multi-constellation receiver array, the proposed algorithm outperforms the traditional baseline-based approach by reducing both high-frequency and low-frequency attitude errors across roll, pitch, and yaw. Despite these promising results, further validation under real high-dynamic scenarios and diverse environmental conditions is required to establish the method’s general applicability. The primary objective of this contribution is not to present a single, universal algorithm. Instead, it is to introduce a problem statement that encourages the development of a family of attitude estimation algorithms and to demonstrate the viability of this approach.

In systems that integrate inertial sensors, pitch and roll angles are typically measured with much higher accuracy (often within a few tenths of a degree) than GNSS-only methods can provide. However, GNSS-based techniques remain particularly valuable for yaw estimation, where lower-grade inertial sensors generally produce larger errors. One promising extension is the use of parallel filtering over multiple yaw a priori values; while computationally demanding, this approach offers near-certain convergence and greater robustness.

The proposed attitude tracking algorithm operates in a recurrent manner, as it provides the attitude estimate at a given epoch based on all estimates from previous epochs. Nevertheless, it can also be adapted for instantaneous, single-epoch attitude determination. To achieve this, a coarse estimate of attitude must be generated at each epoch, which is then refined by solving problem (Equation 24). Taking the idea further, an iterative algorithm operating within a single epoch can likewise be developed. Importantly, convergence properties of the proposed algorithm have not yet been rigorously established. Addressing this question remains an open topic for future investigation.

## 5. Conclusions

In this work, we have formulated a novel problem statement for the attitude estimation problem using GNSS measurements. Based on this formulation, we have developed a GNSS-based attitude tracking algorithm, which is numerically more effective than algorithms of comparable accuracy. The accuracy and computation time of the proposed method were evaluated in a real quasi-static experiment employing a multi-frequency, multi-antenna, multi-GNSS setup. The obtained results were compared with those of a representative classical method based on Wahba’s problem for estimated baselines. Our results demonstrate that given an a priori initial attitude approximation, the proposed method outperforms the traditional baseline-based approach by reducing both high-frequency and low-frequency attitude errors across roll, pitch, and yaw up to 50% without increasing computation time. Simulation results demonstrate that the proposed algorithm performs effectively for short baselines under dynamic conditions.

## Figures and Tables

**Figure 1 sensors-25-06761-f001:**
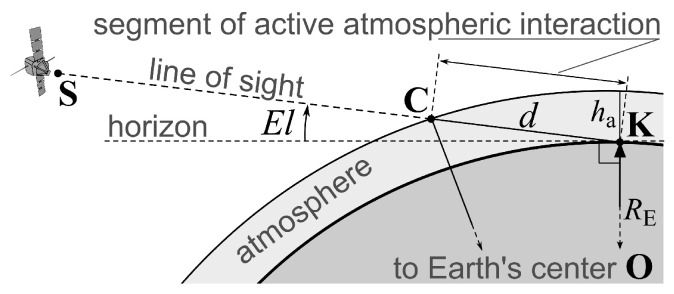
A segment of length *d* of radio signal path from satellite **S** to antenna **K** crossing a nominal atmosphere boundary at point **C** in relation to the elevation angle El with **O** at the Earth’s center.

**Figure 2 sensors-25-06761-f002:**
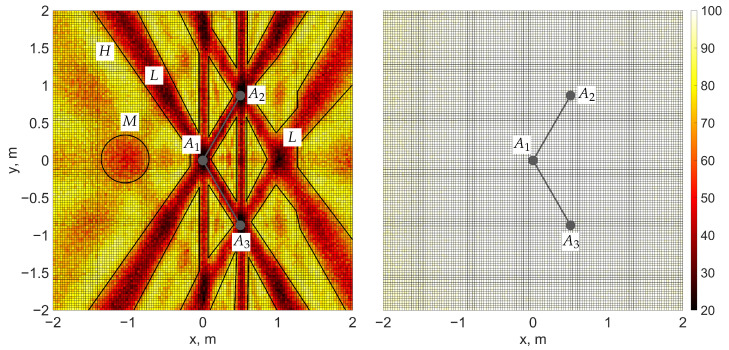
Heatmaps showing integer ambiguity resolution (IAR) success rate (SR) for placing a fourth antenna to form a third baseline with fixed antennas $A_{1}$, $A_{2}$, and $A_{3}$. (**Left**): MC-LAMBDA method; (**right**): proposed method. Regions of low SR (marked by *L*) correspond to parallel or nearly duplicate baselines, which should be avoided. Less obviously, there exists a region of medium SR denoted by *M*, where we choose to place the fourth antenna, so that the IAR process would be more sensitive to changes in its parameters, in contrast to the high SR region (*H*). Each grid point represents an average over 200 iterations with random attitudes, with standard deviations of code and attitude measurements of 20 cm and 3° for MC-LAMBDA and the proposed method, respectively.

**Figure 3 sensors-25-06761-f003:**
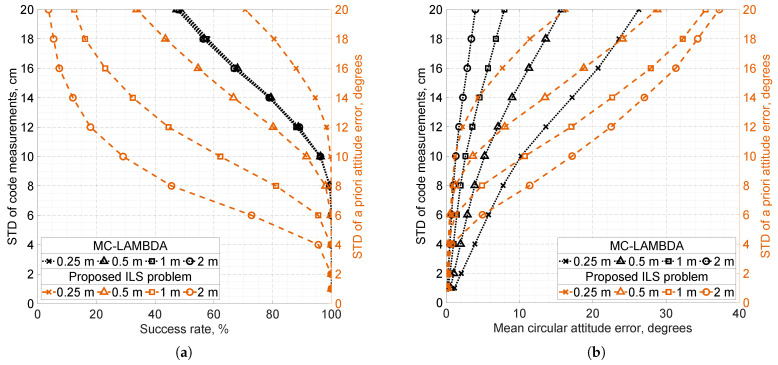
Simulation results for the chosen four-antenna, single-frequency GNSS setup under challenging conditions, comparing the MLAMBDA and MC-LAMBDA methods applied to the proposed and orthonormality-constrained attitude models, respectively. Black dotted lines denote accuracy metrics for MC-LAMBDA, and orange dashed lines for the proposed method. The left vertical axes correspond to the standard deviations (STD) of undifferenced GNSS code measurements, and the right vertical axes to the standard deviations of a priori attitude measurements $\sigma_{\delta }$. Antenna configurations with different baseline lengths are indicated by distinct markers. Each point corresponds to $10^{4}$ independent iterations over a random attitude. (**a**) Success rates of integer ambiguity resolution. (**b**) Mean circular attitude errors.

**Figure 4 sensors-25-06761-f004:**
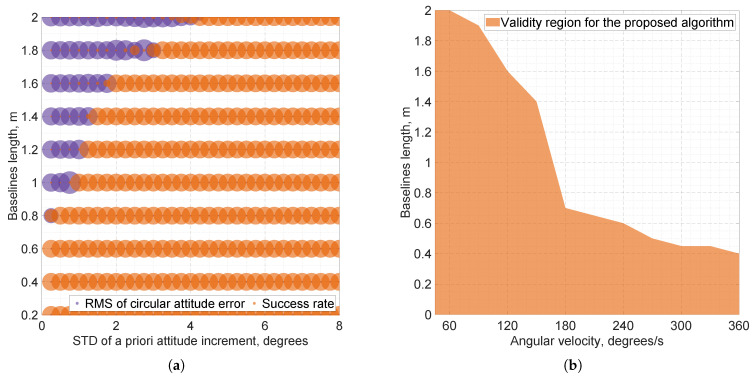
Simulation results of the proposed attitude tracking algorithm for the selected four-antenna, single-frequency GNSS setup under GNSS-challenging conditions and uniform rotations with constant angular velocities. (**a**) Overlapped bubble charts showing success rates of IAR (orange) and root mean square (RMS) circular attitude errors (purple) over varying baseline lengths and a priori standard deviation of attitude increment, $\sigma_{\delta }$, for angular velocity 120°/s. Large purple bubbles correspond to a loss of attitude tracking capability. (**b**) Conservative estimate of the range of applicability for the proposed attitude tracking algorithm.

**Figure 5 sensors-25-06761-f005:**
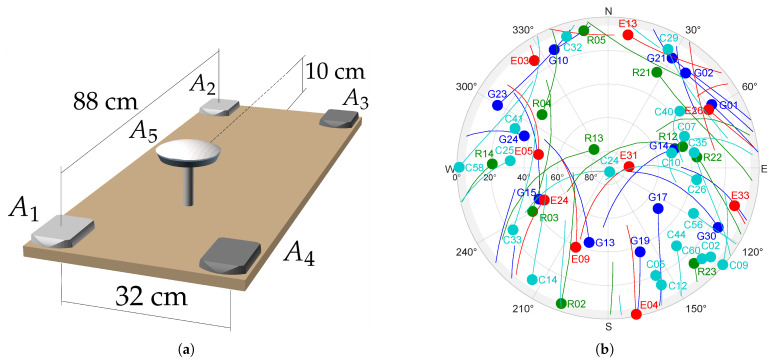
(**a**) Scheme of the multi-antenna GNSS setup in the experiment. (**b**) Tracked satellites on the sky plot seen from 55.7525° latitude, 37.4422° longitude. Here, G, R, E, and C denote GPS, GLONASS, Galileo, and BeiDou systems, respectively.

**Figure 6 sensors-25-06761-f006:**
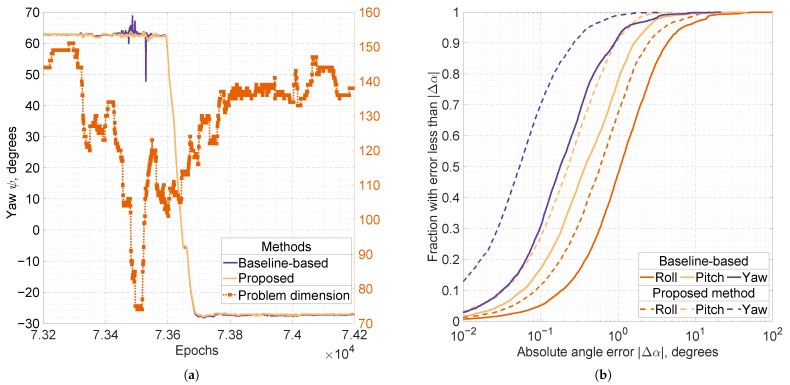
(**a**) Solid lines show the estimates of the yaw angle, ψ, obtained by the baseline-based and proposed methods using the 5-antenna, dual-frequency GNSS setup (GPS, GLONASS, Galileo, and BeiDou). The dotted line with square markers represents the problem dimension for the proposed method, illustrating its computational complexity. (**b**) Cumulative distribution functions of the absolute angle errors, |Δα|, produced by the baseline-based (solid lines) and proposed (dashed lines) methods. Distributions are computed over epochs corresponding only to rotation intervals in the experiment.

**Figure 7 sensors-25-06761-f007:**
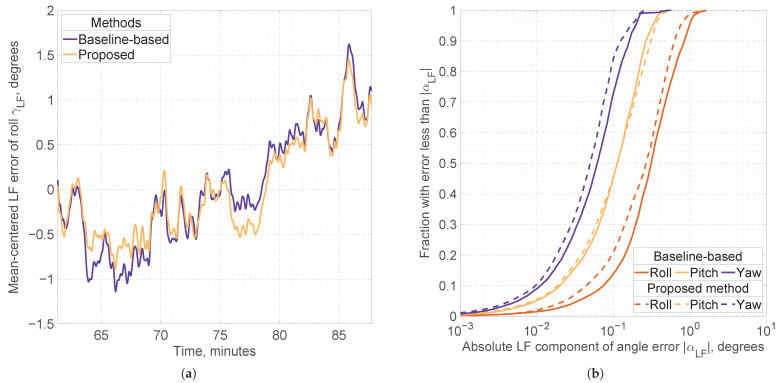
(**a**) Mean-centered low-frequency errors of the roll angle, $\gamma_{\mathrm{LF}}$, obtained by the baseline-based and proposed methods using the 5-antenna, dual-frequency GNSS setup (GPS, GLONASS, Galileo, and BeiDou) during one of the static intervals. (**b**) Cumulative distribution functions of the absolute low-frequency angle errors, $|\Delta \alpha_{\mathrm{LF}}|$, produced by the baseline-based (solid lines) and proposed (dashed lines) methods. Distributions are computed over epochs corresponding only to static intervals in the experiment.

**Table 1 sensors-25-06761-t001:** Simulation setup parameters.

Parameter	Value(s)
GNSS carrier wavelength (frequency)	19.029 cm (L1)
Number of satellites (PDOP)	5 (2.5)
Satellite azimuth and elevation angles, °	(0,17), (−90,24), (−180,31), (90,37), (34,70)
St. dev. of undifferenced phase noise	0.01 cycles ≈ 2 mm
Number of baselines	3
Baseline lengths	0.2–2 m
St. dev. of undifferenced code noise	1–20 cm
St. dev. of a priori attitude error, $\sigma_{\delta }$	1–20°

**Table 2 sensors-25-06761-t002:** Simulation setup in dynamic experiments with uniform rotations.

Parameter	Value(s)
GNSS frequency	20 Hz
Experiment time	100 s
Angular velocities	30–360°/s

**Table 3 sensors-25-06761-t003:** GNSS hardware used in the experiment and the corresponding signal types. The first column lists the short names of the antennas. Carrier phase measurement signals for different GNSS constellations are given in RINEX 3 format. For brevity, L7{I, D} denotes both L7I and L7D signals.

	Antenna	Receiver	Signal Types			
			GPS	GLONASS	Galileo	BeiDou
$A_{1}$	Antcom 743GNSSA (Antcom Corporation, Torrance, CA, USA)	NovAtel PwrPak7D	L1C, L2W	L1C, L2P	L1C, L7Q	L2I, L7{I, D}
$A_{2}$	Antcom 743GNSSA					L2I
$A_{3}$	Javad AirAnt (Javad GNSS, San Jose, CA, USA)	Javad Prego	L1C, L2W	L1C, L2P	–	–
$A_{4}$	Javad AirAnt	Javad Prego	L1C, L2W	L1C, L2P	–	–
$A_{5}$	NovAtel GNSS-804	NovAtel PwrPak7	L1C, L2W	L1C, L2P	L1C, L7Q	L2I, L7{I, D}

**Table 4 sensors-25-06761-t004:** Standard deviations of high-frequency components of attitude angles obtained using two multi-antenna configurations and two attitude estimation methods. Statistics are computed over time intervals corresponding to static periods. G, R, E, and C denote GPS, GLONASS, Galileo, and BeiDou systems, respectively.

Configuration	Method	$\sigma (\gamma_{\mathrm{HF}})$, °	$\sigma (\theta_{\mathrm{HF}})$, °	$\sigma (\psi_{\mathrm{HF}})$, °
3 ant., G, L1	Baseline-based	1.10	0.47	0.15
	Proposed	0.61	0.22	0.10
5 ant., GREC, L1+L2	Baseline-based	0.39	0.13	0.08
	Proposed	0.22	0.10	0.04

**Table 5 sensors-25-06761-t005:** Fraction of epochs with outliers in attitude angles obtained using two multi-antenna configurations and two attitude estimation methods. Statistics are computed over all epochs of the experiment. G, R, E, and C denote GPS, GLONASS, Galileo, and BeiDou systems, respectively.

Configuration	Method	Roll Outliers, %	Pitch Outliers, %	Yaw Outliers, %
3 ant., G, L1	Baseline-based	21.1	21.0	27.6
	Proposed	6.4	12.8	5.1
5 ant., GREC, L1+L2	Baseline-based	2.9	2.8	1.6
	Proposed	1.8	1.8	0.8

**Table 6 sensors-25-06761-t006:** The 99.7-th percentiles of the absolute values of low-frequency components of attitude angles obtained using two multi-antenna configurations and two attitude estimation methods. Statistics are computed over time intervals corresponding to static periods. G, R, E, and C denote GPS, GLONASS, Galileo, and BeiDou systems, respectively.

Configuration	Method	$P_{99.7}\left(\right|\gamma_{\mathrm{LF}}\left|\right)$, °	$P_{99.7}\left(\right|\theta_{\mathrm{LF}}\left|\right)$, °	$P_{99.7}\left(\right|\psi_{\mathrm{LF}}\left|\right)$, °
3 ant., G, L1	Baseline-based	4.53	1.89	1.18
	Proposed	2.60	1.26	0.43
5 ant., GREC, L1+L2	Baseline-based	1.50	0.46	0.43
	Proposed	1.31	0.50	0.25

**Table 7 sensors-25-06761-t007:** Statistics of computation times for two attitude estimation methods. The statistics are computed over time intervals corresponding to static periods. G, R, E, and C denote GPS, GLONASS, Galileo, and BeiDou systems, respectively. MAD refers to the median absolute deviation.

Configuration	Method	Median Time, s	MAD of Time, s
5 ant., GREC, L1+L2	MC-LAMBDA	1.80	0.80
	Proposed	0.011	0.002

## Data Availability

The GNSS broadcast ephemerides used in this study were obtained from the Information and Analysis Center for Positioning, Navigation and Timing (IAC PNT) FTP archive (ftp://ftp.glonass-iac.ru/MCC/BRDC/2023, accessed on 28 October 2025), specifically the files Brdc2140.23f, Brdc2140.23g, Brdc2140.23l, and Brdc2140.23n. The GNSS measurements in CompactRINEX format are provided in the Appendix A, accompanied by detailed descriptions of the data and experimental setup. For more information, please refer to the readme.txt file included in the archive.

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
