# Peer review of "Attitude Tracking Algorithm Using GNSS Measurements from Short Baselines"

_sensors, 2025, doi:10.3390/s25216761_

Round 1
Reviewer 1 Report
Comments and Suggestions for Authors
This is a paper on an important topic, namely that of GNSS-based attitude determination. The authors present a novel GNSS-based attitude determination method that significantly reduces computational complexity without compromising the accuracy achieved by established algorithms. Also, the authors introduce an a priori error model for GNSS measurement errors that lends itself to a clear and intuitive geometric interpretation. In my opinion, this paper is not of high enough interest or significance for inclusion in Sensors.
The paper reduces the nonlinear orthogonality-constrained attitude determination problem to a linear integer least squares problem. The proposed method of attitude determination using carrier phase measurements is based on the idea of simplification of the problem (11). However, the authors should prove that the success rate of the entire week's ambiguity resolution is not affected by this approximation. The algorithm statement is too abstract and the author should provide specific examples for equation (16)-(20). I cannot follow them, please give the analysis for the assumption that the initial attitude is approximately known within a few-degree error.
The statistical properties for so-called regularizing pseudo measurements in equation (21) should be conducted with strict theoretical deduction.
Line 186: Let the attitude at a fixed epoch be approximately estimated. This assumption is not very reasonable in practice for instantaneous GNSS-based attitude determination. If a widely adopted sensor combination employs an inertial measurement unit (IMU) with GNSS data is utilized, the title of the paper should be revised focus on the ‘GNSS/INS Integration Techniques’ and theoretical analysis should be conducted on the success rate of integer ambiguity resolution based on the degree of approximation.
In my opinion, the author's method is essentially just a supplement to MC-LAMBDA with the assumption that the initial attitude is approximately known. This method lacks substantial innovation and is difficult to apply in practice. The core issue that should be focused on is the high success rate instantaneous integer ambiguity resolution, which does not rely on any known information.
There are also few references on attitude determination using GNSS/IMU measurements.
Author Response
This is a paper on an important topic, namely that of GNSS-based attitude determination. The authors present a novel GNSS-based attitude determination method that significantly reduces computational complexity without compromising the accuracy achieved by established algorithms. Also, the authors introduce an a priori error model for GNSS measurement errors that lends itself to a clear and intuitive geometric interpretation. In my opinion, this paper is not of high enough interest or significance for inclusion in Sensors.
Comments 1: The paper reduces the nonlinear orthogonality-constrained attitude determination problem to a linear integer least squares problem. The proposed method of attitude determination using carrier phase measurements is based on the idea of simplification of the problem (11). However, the authors should prove that the success rate of the entire week's ambiguity resolution is not affected by this approximation.
Response 1: We appreciate your feedback. Since an analytical comparison of the ambiguity resolution success rates for problem (24) and the MC-LAMBDA method is a non-trivial task, we have included simulation results to numerically compare the success rates of the proposed problem (24) and the MC-LAMBDA method in Sections 3.1, 3.2.1.
Comments 2: The algorithm statement is too abstract and the author should provide specific examples for equation (16)-(20). I cannot follow them, please give the analysis for the assumption that the initial attitude is approximately known within a few-degree error.
Response 2: A more detailed discussion of the methods for estimating R' was provided in lines 243-262. To address your concern and enhance clarity, we have added a few explanatory sentences following Eq. (17), including specific examples concerning the assumption that the initial attitude is approximately known within a few degrees of error.
Comments 3: The statistical properties for so-called regularizing pseudo measurements in equation (21) should be conducted with strict theoretical deduction.
Response 3: Thank you for your valuable remark. In response, we have reconsidered the terminology and replaced “regularizing pseudo measurements” with the more intuitive and physically meaningful terms “attitude error measurements” and “attitude increment measurements.” While mathematically these measurements can be viewed as regularizing, the key difference is that attitude increment measurements δ are directly derived from known information about the attitude matrix R'. Unlike traditional regularization techniques that impose constraints on state components to address ill-posed problems, our approach does not enforce unnatural assumptions on the state vector.
Regarding the statistical properties, if the estimate R' is biased, the zero-mean assumption for the measurement error E[r] = 0 may not strictly hold. Similar violations occur in the presence of outliers in code or phase measurements. However, since the attitude increment δ is small by definition (||δ|| ≪ 1 rad), it is reasonable to maintain the zero-mean assumption because any bias in R' is expected to be minimal. Otherwise, a significantly biased R' would invalidate our core assumptions and the underlying model. We have added the corresponding simulations in Sections 3.1, 3.2.
Comments 4: Line 186: Let the attitude at a fixed epoch be approximately estimated. This assumption is not very reasonable in practice for instantaneous GNSS-based attitude determination. If a widely adopted sensor combination employs an inertial measurement unit (IMU) with GNSS data is utilized, the title of the paper should be revised focus on the ‘GNSS/INS Integration Techniques’ and theoretical analysis should be conducted on the success rate of integer ambiguity resolution based on the degree of approximation.
Response 4: Thank you for the remark. Our work does not assume the explicit use of an IMU. All presented experimental results obtained using only GNSS-code and carrier phase measurements. The proposed attitude tracking algorithm can be initialized by any single-epoch GNSS-based attitude determination method, as noted below Algorithm 1. You are absolutely correct that the assumption of approximately known attitude parameters is not commonly met in instantaneous GNSS-based attitude determination problems. To avoid misinterpretation and manage expectations, we have renamed the paper to “Attitude Tracking Algorithm using GNSS Measurements from Short Baselines.”
In addition, another useful way to utilize the proposed method is to refine and correct attitude estimates obtained by other attitude determination algorithms. This complements existing methods by improving their accuracy and robustness through continuous tracking and updating.
The proposed algorithm is not intended to be a method for single-epoch instantaneous GNSS-based attitude determination, although it could potentially be developed and adapted for that purpose in the future using ideas, as discussed in the manuscript. Regarding the study of the approximation’s influence on integer ambiguity resolution, a simulation has been conducted and reported in Section 3.2.
Comments 5: In my opinion, the author's method is essentially just a supplement to MC-LAMBDA with the assumption that the initial attitude is approximately known. This method lacks substantial innovation and is difficult to apply in practice. The core issue that should be focused on is the high success rate instantaneous integer ambiguity resolution, which does not rely on any known information.
Response 5: Thank you for this comment. While it is true that our method leverages an initial approximate attitude estimate, it extends beyond a simple supplement to MC-LAMBDA by providing an attitude tracking framework. This tracking capability significantly enhances practical applicability in scenarios where instantaneous methods may face challenges.
For example, when the number of available GNSS measurements is substantially reduced within a single epoch, such that M < 4, instantaneous single-epoch algorithms may fail to yield a solution. In contrast, the proposed method is capable of handling such degradation and can continue to track the attitude correctly for some time thereafter.
The implementation of the MC-LAMBDA method in real-time applications remains a considerable challenge. Let us assume that the proposed algorithm can be (re)initialized with attitude obtained from an instantaneous attitude determination method. Under this assumption, the proposed tracking strategy substantially reduces the frequency of computationally demanding reinitializations, restricting them to the initial initialization stage, to periods following extended GNSS signal outages, or to instances of rapid angular motion of the navigation object. This approach greatly reduces the computational load during most epochs. We have added a computation time comparison in Section 3.4.4.
We have also emphasized the difference between MC-LAMBDA and the methods based on problem (24) in the paragraph beginning with “The key difference between the method based on problem…” in Section 2.3.
Comments 6: There are also few references on attitude determination using GNSS/IMU measurements.
Response 6: We appreciate the comment. Since the proposed algorithm does not necessarily imply the use of an IMU at every epoch (or all epochs), and IMU/GNSS integration is a broad topic, the article initially included only a few references to IMU/GNSS attitude determination. We have now added three more relevant references on IMU/GNSS attitude determination in the sentence: “A widely adopted sensor combination employs an IMU with GNSS data...” in Section 2.3.
Reviewer 2 Report
Comments and Suggestions for Authors
This paper presents a novel method for GNSS-based attitude determination. The authors also introduce a physically-motivated a priori error model for GNSS measurements. The method is validated using a quasi-static experiment with a multi-antenna setup, comparing its performance against a traditional baseline-based approach. The paper is well-structured and the proposed method is mathematically sound and interesting. However, there are several major points that need to be addressed to strengthen the paper's claims and improve its overall quality.
- The authors motivate their work by highlighting the high computational cost of methods like MC-LAMBDA, which solve the full orthonormality-constrained problem. They claim their method offers "accuracy comparable to the MC-LAMBDA method but with significantly lower computational complexity" (line 97-98). However, this claim is never substantiated with evidence. The entire experimental section compares the proposed algorithm only against the "baseline-based method," which is known to be less accurate as it discards orthonormality constraints during the IAR step. To validate the central claims of the paper, the authors must include a direct comparison with the MC-LAMBDA method.
- The experiment was conducted under "quasi-static" conditions, involving long static periods and slow, controlled rotations. While this is a good starting point, it does not demonstrate the algorithm's robustness in more realistic, dynamic scenarios that are common in navigation applications. The authors should include simulations or experiments under dynamic conditions, such as with angular motion greater than 120°/s, to validate robustness.
- In the Discussion, the authors correctly state that the convergence properties of an iterative version of the algorithm have not been established. While a full proof is beyond the scope of this paper, could the authors comment on the conditions under which they expect the algorithm to converge?
- The a priori variance model in Section 2.2 is a nice contribution, leading to ha = 320 km. This value for the "nominal total thickness" of "actively interacting atmospheric layers" seems quite large. Could the authors clarify what atmospheric phenomena that this ha is intended to represent? A little more physical context would be beneficial.
Author Response
This paper presents a novel method for GNSS-based attitude determination. The authors also introduce a physically-motivated a priori error model for GNSS measurements. The method is validated using a quasi-static experiment with a multi-antenna setup, comparing its performance against a traditional baseline-based approach. The paper is well-structured and the proposed method is mathematically sound and interesting. However, there are several major points that need to be addressed to strengthen the paper's claims and improve its overall quality.
Comments 1: - The authors motivate their work by highlighting the high computational cost of methods like MC-LAMBDA, which solve the full orthonormality-constrained problem. They claim their method offers "accuracy comparable to the MC-LAMBDA method but with significantly lower computational complexity" (line 97-98). However, this claim is never substantiated with evidence. The entire experimental section compares the proposed algorithm only against the "baseline-based method," which is known to be less accurate as it discards orthonormality constraints during the IAR step. To validate the central claims of the paper, the authors must include a direct comparison with the MC-LAMBDA method.
Response 1: Thank you for your careful review and insightful comments. You are absolutely right; we did not previously provide explicit arguments to support this statement.
The first argument in favor of the proposed attitude tracking method based on Eq. (24) achieving accuracy comparable to MC-LAMBDA is that both approaches utilize the orthogonality-constrained attitude model and are consistent with the formulation in problem (11). Furthermore, the method based on solving problem (24) enforces the orthonormality constraints on the attitude matrix at every step of the integer ambiguity resolution procedure, even more strictly than MC-LAMBDA. To clarify this point, we have added a dedicated paragraph beginning with “The key difference between the method based on problem…” to Section 2.3.
The main reason for comparing the accuracy of the proposed method with baseline-based approach is the similarity in their respective computational complexities and average computation times. In both cases, the majority of the computational effort arises from solving integer least-squares (ILS) problems of similar dimensions: 3*B+M for the baseline-based method and 3+M for the proposed algorithm.
To directly address your concern, we have also included a comparison between the proposed method and the MC-LAMBDA approach, considering both accuracy and computation times. The comparison based on simulations is presented in Sections 3.1 and 3.2. Additionally, Section 3.4.4 now contains a computational time analysis derived from our quasi-static real experiment.
Comments 2: - The experiment was conducted under "quasi-static" conditions, involving long static periods and slow, controlled rotations. While this is a good starting point, it does not demonstrate the algorithm's robustness in more realistic, dynamic scenarios that are common in navigation applications. The authors should include simulations or experiments under dynamic conditions, such as with angular motion greater than 120°/s, to validate robustness.
Response 2: We appreciate this feedback. Following your advice, we have added research dedicated to estimating the applicability range of the proposed method under dynamic conditions in Sections 3.1 and 3.2, specifically in Section 3.2.2. Additionally, to better clarify the range of validity of the method, we have renamed the manuscript to “Attitude Tracking Algorithm using GNSS Measurements from Short Baselines.”
Comments 3: - In the Discussion, the authors correctly state that the convergence properties of an iterative version of the algorithm have not been established. While a full proof is beyond the scope of this paper, could the authors comment on the conditions under which they expect the algorithm to converge?
Response 3: The key assumption underlying our attitude tracking algorithm is that an approximate attitude matrix is already known from the previous epoch, i.e., the attitude increment δ is sufficiently small. If this assumption is violated, one cannot expect the algorithm to exhibit favorable convergence properties. An interesting direction for future research would be to estimate the size of the convergence region as a function of δ and under varying conditions of experiment and algorithm tuning parameters. A better understanding of this region would allow for more confident and precise predictions regarding the applicability of this version of the proposed algorithm than is currently possible. Indirectly we investigated this issue in the Section 3.2.1, where region of high-accuracy was obtained for short baselines.
Comments 4: - The a priori variance model in Section 2.2 is a nice contribution, leading to ha = 320 km. This value for the "nominal total thickness" of "actively interacting atmospheric layers" seems quite large. Could the authors clarify what atmospheric phenomena that this ha is intended to represent? A little more physical context would be beneficial.
Response 4: You are right, thank you for the observation. After Eq. (15), we have added a clarification and a reference to one of the classic books on atmospheric electrodynamics showing upper layer of maximum electron density region to be at roughly 300 km altitude.
Reviewer 3 Report
Comments and Suggestions for Authors
The presented manuscript offers a new algorithm for determining orientation using phase measurements obtained using multiple antennas of Global Navigation Satellite Systems (GNSS). The proposed algorithm is based on classical numerical techniques for solving the integer least squares (ILS) problem and does not require computationally intensive numerical methods such as MC-LAMBDA, which deals with non-linear constraints and a non-convex search space. However, the proposed algorithm requires an initial approximation of the attitude, so it operates in a recurrent manner. It provides the attitude estimate at a given epoch based on all estimates from previous epochs. The principal advantage of the proposed approach is reduced computational effort in resolving the ILS problem.
The paper presents the test results of an experimental setup involving five different multi-constellation receiver arrays. The results obtained showed what proposed algorithm outperforms traditional baseline-based approach by reducing both high-frequency and low-frequency attitude errors in static conditions for high-rate GNSS observations. Further validation under dynamic scenarios and diverse environmental conditions is required to establish the method’s general applicability.
The main advantage of the proposed algorithm, as stated in the paper, is a much more computationally efficient solution. However, the paper lacks any qualitative or quantitative assessment of the computational complexity of traditional and proposed attitude determination algorithms. It is necessary to provide calculations confirming the computational efficiency of the proposed algorithm.
The data presented in Table 3 differs from their description presented in lines 380-383 and require clarification.
Author Response
The presented manuscript offers a new algorithm for determining orientation using phase measurements obtained using multiple antennas of Global Navigation Satellite Systems (GNSS). The proposed algorithm is based on classical numerical techniques for solving the integer least squares (ILS) problem and does not require computationally intensive numerical methods such as MC-LAMBDA, which deals with non-linear constraints and a non-convex search space. However, the proposed algorithm requires an initial approximation of the attitude, so it operates in a recurrent manner. It provides the attitude estimate at a given epoch based on all estimates from previous epochs. The principal advantage of the proposed approach is reduced computational effort in resolving the ILS problem.
The paper presents the test results of an experimental setup involving five different multi-constellation receiver arrays. The results obtained showed what proposed algorithm outperforms traditional baseline-based approach by reducing both high-frequency and low-frequency attitude errors in static conditions for high-rate GNSS observations. Further validation under dynamic scenarios and diverse environmental conditions is required to establish the method’s general applicability.
Comments 1: The main advantage of the proposed algorithm, as stated in the paper, is a much more computationally efficient solution. However, the paper lacks any qualitative or quantitative assessment of the computational complexity of traditional and proposed attitude determination algorithms. It is necessary to provide calculations confirming the computational efficiency of the proposed algorithm.
Response 1: Thank you for your review. We agree with your remark. Given that analytically estimating the computational time or complexity of algorithms such as MC-LAMBDA is a challenging and non-trivial task, we have instead focused on comparing computational complexities through empirical experiments. First, we have added additional context regarding typical computation times for MC-LAMBDA and the ILS problem solution via LAMBDA, as reported by the original authors of the MC-LAMBDA method, in Section 2.3 (lines 252-259). Second, we conducted a comparative analysis of computation times between the proposed attitude tracking algorithm and MC-LAMBDA using our experimental real data and an implementation of MC-LAMBDA. The corresponding results are presented in Section 3.4.4, Table 7.
Comments 2: The data presented in Table 3 differs from their description presented in lines 380-383 and require clarification.
Response 2: Thank you for pointing out this issue. We have carefully rephrased the two sentences after Table 5 to ensure that their description now does not contradict the data presented in the table.
Reviewer 4 Report
Comments and Suggestions for Authors
In the peer-reviewed article propose a new method that reduces the nonlinear orthogonality-constrained attitude determination problem to a linear integer least squares problem, for which numerical methods are well known and computationally much less demanding. Additionally, a simple a priori model for Global Navigation Satellite Systems (GNSS) measurement error variance is introduced, grounded on the geometry of satellite signal propagation through vacuum and the Earth’s atmosphere, providing a clear physical interpretation.
The problem solvable in the manuscript is relevant.
This manuscript by the authors is a continuation of long-term studies in the area under consideration.
Because the primary objective of this contribution is not to present a single, universal algorithm, it is to introduce a problem statement that encourage the development of a family of attitude determination algorithms and to demonstrate viability of this approach, then I will omit a number of comments. Also, some of my suggestions coincide with those indicated by the authors as future research in section 4.
Overall, the research seems interesting and the article is written at a good level.
The literature review is sufficient and the references are appropriate.
The conclusions correspond to the presented research results and arguments. There are no fundamental comments on the research.
- The authors write: 'for different combinations of GNSS antennas and receivers, σ0 may vary from approximately 0.005 to 0.03 cycles'. Why do you use the value 0.01 for calculations, and not the average or 0.02, for example? Table A1 also has two values for different antennas. Please describe in more detail.
- The authors write: 'To enable joint processing of measurements from both receiver types, the NovAtel data was decimated to 10 Hz, and all measurements are then utilized at this common sampling rate'. There may be an inaccuracy in the sentence, because it should be 20 Hz?
- Write what the variables 'm' in formula 2 and 'Q' in formula 3 represent.
- In line 148, a symbol in square brackets be missing?
- Are there any limitations or constraints in the algorithm presented in the paper that researchers should be aware of when applying them to practical problems?
Author Response
In the peer-reviewed article propose a new method that reduces the nonlinear orthogonality-constrained attitude determination problem to a linear integer least squares problem, for which numerical methods are well known and computationally much less demanding. Additionally, a simple a priori model for Global Navigation Satellite Systems (GNSS) measurement error variance is introduced, grounded on the geometry of satellite signal propagation through vacuum and the Earth’s atmosphere, providing a clear physical interpretation.
The problem solvable in the manuscript is relevant.
This manuscript by the authors is a continuation of long-term studies in the area under consideration.
Because the primary objective of this contribution is not to present a single, universal algorithm, it is to introduce a problem statement that encourage the development of a family of attitude determination algorithms and to demonstrate viability of this approach, then I will omit a number of comments. Also, some of my suggestions coincide with those indicated by the authors as future research in section 4.
Overall, the research seems interesting and the article is written at a good level.
The literature review is sufficient and the references are appropriate.
The conclusions correspond to the presented research results and arguments. There are no fundamental comments on the research.
Comments 1: The authors write: 'for different combinations of GNSS antennas and receivers, σ0 may vary from approximately 0.005 to 0.03 cycles'. Why do you use the value 0.01 for calculations, and not the average or 0.02, for example? Table A1 also has two values for different antennas. Please describe in more detail.
Response 1: Thank you for the review. After reference to Table A1 in the penultimate paragraph of Section 3.4, we have added sentence, which hopefully clarify our particular choice.
Comments 2: The authors write: 'To enable joint processing of measurements from both receiver types, the NovAtel data was decimated to 10 Hz, and all measurements are then utilized at this common sampling rate'. There may be an inaccuracy in the sentence, because it should be 20 Hz?
Response 2: The minor concern was re-examined, and we confirm that the content is correct. To process measurements from receivers of both types (Javad and NovAtel) we decimated data with higher-frequency sampling rate (20 Hz) from NovAtel receivers to 10 Hz frequency, which is sampling rate of Javad receivers. In other words, processing data from Configuration B, we ignore half of all available measurements from antennas A1, A2, and A5. In Configuration A only Novatel receivers are involved, so we use measurements on 20 Hz sampling frequency.
Comments 3: Write what the variables 'm' in formula 2 and 'Q' in formula 3 represent.
Response 3: Thank you for the remark. We have explicitly clarified the meaning of the scalar “m_i” immediately following Eq. (2). Regarding the covariance matrices, we consider that they are a priori known and defined by Eqs. (3) and (5). Therefore, we have introduced them as “a priori known covariance matrices”.
Comments 4: In line 148, a symbol in square brackets be missing?
Response 4: Thank you for the question. The vector in question was the column vector [1; 1] containing two ones. We have introduced separate design matrices for the code and carrier phase measurements in Eqs. (1) and (2), and accordingly, Eq. (12) has been revised to prevent potential misinterpretations. As a result, Eqs. (18), (19), (20), (22), and (23) now include the matrices H and H^i with the subscript “ϕ”, with the original “H” and “H^i” no longer appearing.
Comments 5: Are there any limitations or constraints in the algorithm presented in the paper that researchers should be aware of when applying them to practical problems?
Response 5: Thank you for your valuable question. We discuss the main limitations of the proposed algorithm immediately following the introduction of Algorithm 1 in Section 2.3. The key underlying assumption of our method is the smallness of the attitude increment δ, as indicated by Eq. (16). This assumption defines the scope of applicability and the limitations of the algorithm. Simulation results that help determine the applicability range of the proposed method are presented in Sections 3.1 and 3.2. Notably, shorter baselines and higher GNSS sampling rates expand the applicability scope.
Round 2
Reviewer 1 Report
Comments and Suggestions for Authors
The authors have improved the paper's presentation and supplemented relevant data. However, to enable effective reproduction of the algorithm in practical industrial scenarios and achieve robust results, the authors should add a paragraph emphasizing:
(1) the prerequisites, configurations, and required known constraint information for using the algorithm;
(2) the factors limiting the integer ambiguity resolution success rate and the recommended configurations;
(3) the advancements and limitations of the proposed framework compared to the classical MC-LAMBDA algorithm.
Only after the aforementioned issues are clearly addressed and properly resolved can the paper be recommended for publication.
Author Response
Comments 1: The authors have improved the paper's presentation and supplemented relevant data. However, to enable effective reproduction of the algorithm in practical industrial scenarios and achieve robust results, the authors should add a paragraph emphasizing:
(1) the prerequisites, configurations, and required known constraint information for using the algorithm;
(2) the factors limiting the integer ambiguity resolution success rate and the recommended configurations;
(3) the advancements and limitations of the proposed framework compared to the classical MC-LAMBDA algorithm.
Only after the aforementioned issues are clearly addressed and properly resolved can the paper be recommended for publication.
Response 1: Thank you for your revision and valuable remarks. We have added a dedicated paragraph in the first paragraph of the Discussion section addressing the known constraints essential for the effective application of the proposed algorithm. Additionally, we discuss the advancements and limitations of our framework in comparison to the classical MC-LAMBDA algorithm. Alongside these content enhancements, we corrected several grammatical errors and improved Figure 2 to enhance clarity and presentation quality.
Reviewer 3 Report
Comments and Suggestions for Authors
The presented manuscript offers a new algorithm for determining orientation using phase measurements obtained using multiple antennas of Global Navigation Satellite Systems (GNSS). The proposed algorithm is based on classical numerical techniques for solving the integer least squares (ILS) problem and does not require computationally intensive numerical methods such as MC-LAMBDA, which deals with non-linear constraints and a non-convex search space. However, the proposed algorithm requires an initial approximation of the attitude, so it operates in a recurrent manner. It provides the attitude estimate at a given epoch based on all estimates from previous epochs. The principal advantage of the proposed approach is reduced computational effort in resolving the ILS problem.
The paper presents the test results of an experimental setup involving five different multi-constellation receiver arrays. The results obtained showed what proposed algorithm outperforms traditional baseline-based approach by reducing both high-frequency and low-frequency attitude errors in static conditions for high-rate GNSS observations. Further validation under dynamic scenarios and diverse environmental conditions is required to establish the method’s general applicability.
All comments are taken into account.
Author Response
Comments 1: The presented manuscript offers a new algorithm for determining orientation using phase measurements obtained using multiple antennas of Global Navigation Satellite Systems (GNSS). The proposed algorithm is based on classical numerical techniques for solving the integer least squares (ILS) problem and does not require computationally intensive numerical methods such as MC-LAMBDA, which deals with non-linear constraints and a non-convex search space. However, the proposed algorithm requires an initial approximation of the attitude, so it operates in a recurrent manner. It provides the attitude estimate at a given epoch based on all estimates from previous epochs. The principal advantage of the proposed approach is reduced computational effort in resolving the ILS problem.
The paper presents the test results of an experimental setup involving five different multi-constellation receiver arrays. The results obtained showed what proposed algorithm outperforms traditional baseline-based approach by reducing both high-frequency and low-frequency attitude errors in static conditions for high-rate GNSS observations. Further validation under dynamic scenarios and diverse environmental conditions is required to establish the method’s general applicability.
All comments are taken into account.
Response 1: Thank you for your comments and thorough review. We have added a paragraph in the first paragraph of the Discussion section that addresses the advancements and limitations of our method compared to established algorithms. Additionally, we corrected several grammatical errors and improved Figure 2 to enhance clarity and presentation quality.
Round 3
Reviewer 1 Report
Comments and Suggestions for Authors
Attitude determination and attitude tracking are fundamentally distinct concepts. The current abstract fails to reflect the "attitude tracking" mentioned in the title, as it only describes attitude determination. The author needs to clarify the research focus by explaining the connotation of attitude tracking and strengthen thematic coherence through revised expressions in the abstract, introduction, main body, and experimental sections.
Author Response
Comments 1: Attitude determination and attitude tracking are fundamentally distinct concepts. The current abstract fails to reflect the "attitude tracking" mentioned in the title, as it only describes attitude determination. The author needs to clarify the research focus by explaining the connotation of attitude tracking and strengthen thematic coherence through revised expressions in the abstract, introduction, main body, and experimental sections.
Response 1: Thanks for noticing the ambiguity which has been left in some places after our last revision of the text. On the one hand, attitude tracking can be viewed as a subset or extension of attitude determination. On the other hand, we agree that explicitly distinguishing these concepts in the manuscript will enhance the reader’s understanding of the contribution. Accordingly, we have revised the manuscript to clarify this distinction throughout.
In the abstract, we replaced “We propose a new method that reduces…” with “Given an a priori initial attitude approximation, we propose a new algorithm for attitude tracking based on the reduction of…”, to better emphasize the focus on tracking. In the penultimate paragraph of the Introduction, we added a sentence explicitly explaining the difference between attitude determination and attitude tracking as used in this paper. In the Conclusions, we replaced “Our results demonstrate the proposed method…” with “Our results demonstrate, that given an a priori initial attitude approximation, the proposed method…” to emphasize the limitation of the algorithm and distinguish between the concepts.
Moreover, in the captions of Tables 4, 5, 6, and 7, as well as related sections, where both attitude determination and attitude tracking algorithms are being compared, we replaced “…attitude determination methods” with “…attitude estimation methods” to appropriately bridge the concepts and avoid possible misunderstandings. Similarly, in the last paragraph of the Introduction, the second paragraph of the Discussion, and the first paragraph of the Conclusions, we replaced “attitude determination” with “attitude estimation”.
As a result of this and previous revisions, we have replaced “attitude determination” with “attitude tracking” in many contexts, e.g., in lines 4, 11, 21, 96, 99, 107, 188, 189, 353, 356, 401, 570, 601, and 612. Thus, in the latest version of the manuscript, there are no places where our algorithm is referred to as the attitude determination algorithm.
Additionally, we have improved the English for clearer expression of the research.